# Cross-Analysis of mHealth Social Acceptance Among Youth: A Comparative Study Between Japan and China

**DOI:** 10.3390/bs15020213

**Published:** 2025-02-14

**Authors:** Olugbenga Akiogbe, Hanlin Feng, Karin Kurata, Makoto Niwa, Jianfei Cao, Shuo Zhang, Itsuki Kageyama, Yoshiyuki Kobayashi, Yeongjoo Lim, Kota Kodama

**Affiliations:** 1Graduate School of Design and Architecture, Nagoya City University, Nagoya 464-0083, Japan; adeoluakiogbe@gmail.com (O.A.); k.kurata@tsuruoka-nct.ac.jp (K.K.); 2Graduate School of Technology Management, Ritsumeikan University, Ibaraki 567-8570, Japan; feng2402315861@gmail.com (H.F.); cjf869419482@gmail.com (J.C.); 3Research Organization of Open Innovation and Collaboration, Ritsumeikan University, Osaka 567-8570, Japan; mak.niwa@po.nippon-shinyaku.co.jp; 4Institute at School of Management, Harbin Institute of Technology, Harbin 150001, China; 18b310006@stu.hit.edu.cn; 5Merge System Co., Ltd., Fukuoka 810-0041, Japan; kageyama.itsuki@hoshi.ac.jp; 6Medical Data Science Lab., Hoshi University, Tokyo 142-0063, Japan; kobayashi.yoshiyuki@hoshi.ac.jp; 7Graduate School of Corporate Business, Ritsumeikan University, Ibaraki 567-8570, Japan; lim40@fc.ritsumei.ac.jp; 8School of Data Science, Nagoya City University, Nagoya 4467-8501, Japan

**Keywords:** mHealth, social acceptance, youth, Japan, China, comparative path analysis

## Abstract

Although mobile health (mHealth) technologies have emerged as a revolutionary approach to enhance healthcare delivery, few studies have examined how it is perceived and accepted in different cultures. This study investigated mHealth’s social acceptance among young people in Japan and China, with a focus on cultural influences on technology adoption. A comparative analysis approach was adopted, employing an extended unified theory of acceptance and use of the technology model. University students from both countries, recruited using harmonized sampling methods, completed questionnaires. We employed descriptive statistics to summarize the sample characteristics, confirmatory factor analysis to validate the constructs, multigroup analysis to test for measurement invariance and ensure the applicability of the model in both cultural contexts, and comparative path analysis to explore differences in the various factors influencing mHealth acceptance in each cultural setting. The findings revealed distinct cultural effects on mHealth acceptance. Japanese young people showed cautious acceptance influenced by societal norms and infrastructure, whereas Chinese young people demonstrated strong engagement driven by government support and the growing digital health industry. The study emphasizes the importance of considering cultural and systemic dynamics when integrating mHealth into youth healthcare models and suggests tailored strategies for successful implementation.

## 1. Introduction

The emergence and systematic integration of mobile health (mHealth) technologies have had a significant impact on the global healthcare sector ([26]). These technologies offer unique opportunities to enhance healthcare delivery, engage patients, and disseminate health information ([42]). Among young people—often early adopters of technology ([36]), mHealth has the potential to shape health-related behaviors and attitudes that persist into adulthood ([2]).

However, the acceptance and use of mHealth solutions among this demographic vary across different cultural and regional contexts ([86]). mHealth refers to the use of mobile devices such as smartphones and tablets to support healthcare services ([39]). This technology has revolutionized healthcare delivery by enabling remote monitoring, health education, and communication between patients and healthcare providers. mHealth has the potential to improve access to healthcare in remote and underserved areas and empower individuals to take control of their own health. With the surging popularity of mobile devices worldwide, mHealth is becoming an increasingly important tool for improving health outcomes globally ([16]). This study focuses on the social acceptance of mHealth among youth in Japan and China—two countries that differ in their cultural settings, healthcare frameworks, and technology assimilation practices. The study contributes to the existing body of knowledge by providing a cross-cultural analysis on mHealth adoption, aiming to expand the understanding of cultural influences on technology acceptance by applying the extended unified theory of acceptance and use of technology (UTAUT) framework in two distinct healthcare and technological settings. The two countries were selected for their contrasting cultural characteristics and technological capabilities, which make them ideal for a comparative analysis of mHealth adoption. China’s collectivist culture has supported rapid advancements in healthcare technology integration ([46]), reflecting its emphasis on group dynamics and centralized decision-making. Conversely, Japan’s individualistic culture, combined with its aging population and advanced medical infrastructure ([58]), presents unique opportunities and challenges for mHealth adoption. These distinctions provide a rich foundation for exploring how cultural and technological factors shape mHealth acceptance and use among youth populations.

In Japan, the combination of advanced technological infrastructure and a proactive healthcare innovation culture suggests an environment that is seemingly conducive to mHealth adoption. However, Japanese youth’s engagement with applications for managing personal health remains relatively low, suggesting possible cultural and systemic barriers that hinder their broader acceptance and integration ([15]). Their apparent reluctance highlights the need to examine the socio-cultural and systemic dynamics that may influence mHealth adoption among Japanese youth. On the other hand, youth in China exhibit a more robust engagement with mHealth, driven by government advocacy, an emerging digital healthcare industry, and an increasing inclination toward digital solutions for health-related needs ([76]). Young people in China show significant engagement with mHealth apps designed for managing personal health. The influence of social factors plays a critical role in their behavioral intentions to adopt mHealth technologies ([14]). The future impact of mHealth on healthcare depends heavily on its acceptance among young people. Adolescents, who readily embrace new technology, have distinct preferences and behaviors that need to be understood to ensure societal acceptance of mHealth. Social acceptability encompasses more than just user acceptance; it involves wider recognition, endorsement, and integration of mHealth into young people’s lifestyles and social circles. Factors such as ease of use and perceived usability are essential in determining the acceptance of a technology ([18]). However, the opinions of peers and prevailing trends also influence the perceptions of young individuals. The theory of diffusion of innovations, as described by Rogers ([80]), provides a relevant framework for understanding the acceptance of mHealth. It suggests that visibility, trial opportunities, and endorsements from peer networks significantly enhance the rate at which technology is adopted. In the case of mHealth, this means that the benefits and outcomes must be made known to young populations, since peer testimonials and endorsements can greatly encourage its use. Additionally, social media platforms have a substantial influence on the attitudes and feelings of young individuals towards health-related technologies. Platforms that facilitate peer engagement and information exchange play a role in disseminating health information and promoting the widespread use of mHealth ([48]). The participatory nature of these platforms allows for the exchange of mHealth experiences and results, contributing to a community-based validation process that either strengthens or weakens social acceptance ([29]). However, the integration of mHealth into youth healthcare models requires addressing issues of privacy, data security, and trust, which are vital for the widespread acceptance of technology ([87]). Ensuring confidentiality and implementing robust data protection measures are essential for building confidence among young users, which in turn promotes the social acceptance and long-term integration of mHealth solutions. Additionally, the effectiveness and ease of use of mHealth applications are crucial factors in determining their acceptability. Technologies that are user-friendly, engaging, and customized to the preferences and needs of young individuals are more likely to be embraced and advocated within their social circles ([38]). Therefore, involving young individuals in the creation and development of mHealth applications greatly enhances their relevance and attractiveness, leading to improved societal acceptance. Customized educational programs on the benefits and applications of mHealth can enhance its societal acceptance by aligning with youth’s values and communication preferences. By encouraging informed discussions, these programs can clarify misconceptions, address concerns, and highlight the potential of mHealth in empowering young individuals to take control of their health and improve their health literacy.

The acceptance of mHealth among young people depends on various factors, including peer influence, social media dynamics, privacy assurances, functional appeal, and educational outreach. The relationship between technology and healthcare continues to evolve with new advancements. To fully leverage the transformative capabilities of mHealth in young populations and promote its broader implementation and influence in the healthcare system, it is essential to comprehend and actively cultivate these elements of societal acceptance. Despite these insights, the existing literature still lacks a detailed understanding of how differing cultural contexts, such as in Japan and China, specifically impact the acceptance and integration of mHealth among youth. This study aims to fill this gap by providing an empirical comparison of the factors influencing mHealth acceptance in these two distinct cultural settings. The following research questions were addressed to achieve the research aims: (1) What are the key determinants of mHealth acceptance among youth in Japan and China? (2) How do cultural factors influence the attitudes and behaviors of youth towards mHealth adoption?

## 2. Theoretical Framework and Proposed Model

Our study employs an extended UTAUT framework. Aligned with the model used by [15] ([15]), who added additional constructs (health consciousness, trust, and perceived risk), a modified UTAUT model was developed to gain a comprehensive understanding of the factors that influence mHealth technology acceptance. The model comprises eight key constructs that play a crucial role, categorized into four dimensions (Table 1), as reflected in the hypothesis path model. Its focus on detailed psychological and contextual factors seeks to uncover the subtle dynamics of technology acceptance, providing valuable insights for promoting the adoption and engagement of mHealth technology among the target demographics.

Building on this foundation, we hypothesize a bi-directional effect of behavioral intentions and perceptions towards mHealth technologies, constituting a two-tailed approach in our analysis. First, we posit that perceived risk (H1) will have a differential negative impact on the behavioral intentions to use mHealth technologies in both Japan and China. Trust (H2) is hypothesized to positively influence behavioral intentions in both cultural contexts, with an anticipated variability in strength. Similarly, performance expectancy (H3) is expected to positively correlate with behavioral intentions, with potential differences in how strongly it influences each group. Effort expectancy (H4) is also predicted to have a positive influence on behavioral intentions, with a comparative analysis revealing possible disparities in how it is perceived in Japan versus China. Furthermore, health consciousness is hypothesized to influence mHealth perceptions through mediation effects, where perceived risk (H5a), trust (H5b), and perceived usefulness (H5c) mediate the relationship between health consciousness and behavioral intention. These effects are examined within each cultural setting to determine variances. Social influence (H6) is similarly proposed to moderate perceptions of mHealth technologies, specifically, to decrease perceived risk (H6a), increase trust (H6b), and enhance perceived usefulness (H6c), with these influences expected to vary between Japanese and Chinese youth. Lastly, facilitating conditions are anticipated to enhance effort expectancy (H7) concerning the adoption of mHealth technologies, with a focus on identifying divergent influences between the two study groups. This will allow for a deeper understanding of the cultural differences in mHealth technology acceptance and provide insights into the specific factors that drive these variances.

## 3. Materials and Methods

This study is grounded in the principles of comparative analysis—a methodological approach that systematically examines and compares different entities or phenomena to identify similarities and differences ([82]). It is widely employed across various disciplines to foster a deeper comprehension of diverse subjects ([9]; [24]). We used standardized questionnaires (Appendix E) to access young people’s perceptions, usage intentions, and prospective use of mHealth technologies, supplemented by demographic information. The data were collected by independent researchers focused on either country to enable detailed subgroup analyses. Appendix A displays the methodology flowchart representing the steps undertaken to ensure accuracy and reproducibility.

### 3.1. Sampling Characteristics

This study adopted a harmonized sampling approach across educational settings in Japan and China ([49]), focusing on university students. In Japan, this involved survey distribution within a university context, offering students the option to engage with the questionnaire either directly in class via a QR code or through an online academic platform. This approach ensured inclusivity and broad participation, yielding responses from 233 individual students reflecting Japanese youth’s perspectives towards mHealth. In China, a parallel approach, including online platforms in classrooms and links provided through online learning systems, was adopted within similar educational settings, ensuring consistency in the study’s comparative framework by employing similar methods to those used in Japan. This strategy was intended to maximize response rates and engage a technologically connected demographic, crucial for understanding mHealth acceptance in countries with rapidly evolving healthcare technology landscapes. The Chinese segment of the study successfully gathered responses from 337 individual students, enabling a robust cross-cultural analysis when juxtaposed with the Japanese data. By aligning the sampling methods across both countries within educational settings, the study ensured that the data collection was comparable, reliable, and reflective of young adults’ attitudes toward mHealth.

### 3.2. Data Analysis

Data analyses were performed using IBM SPSS version 28.0 ([84]), and AMOS version 28.0 and RStudio for additional statistical computations. Initially, a descriptive analysis was conducted using SPSS to summarize the sample characteristics and provide an overview of the mean and dispersions of the demography variables. Confirmatory factor analysis (CFA) was employed to validate the measurement model ([57]), ensuring construct reliability and validity. Multigroup structural equation modeling (SEM) was employed to test hypothesized relationships and assess latent mean differences ([63]). All analyses involving CFA and SEM were conducted using AMOS software. A post hoc power analysis was conducted to evaluate the adequacy of the sample size for detecting the hypothesized model fit using root mean square error of approximation (RMSEA) = 0.05 as the effect size ([41]; [53]). While post hoc power analysis has been criticized for its reliance on sample-specific parameters ([1]; [64]), it serves as a supplementary tool in our study to assess the robustness of our analysis ([75]; [92]). The total sample comprised 233 participants from Japan and 337 from China (N = 570). The degrees of freedom (df = 693) for the structural equation model were calculated based on the number of observed variables (41) and estimated parameters (168; [17]), derived as follows:df = Total Data Points − Estimated Parameters The total number of data points was computed as $\frac{k\left(k+1\right)}{2}$, where k is the number of observed variables (*k* = 41), resulting in 861 data points. The estimated parameters included factor loadings, error variances, latent variances, covariances, and structural paths, summing to 168. This yielded df = 861 − 168 = 693. Using the semPower package in R ([45]), post hoc power was calculated as 1.00, confirming that the sample size from both groups was sufficient to detect a close model fit at RMSEA = 0.05 with a significance level of α = 0.05 ([71]). To assess the reliability of the constructs, Cronbach’s alpha coefficients were calculated, with an acceptability threshold of 0.70 ([81]). The validity of the measurement model was evaluated by examining the factor loadings, average variance extracted (AVE), and composite reliability (CR) statistics, adhering to the recommended cut-off values of 0.50 for AVE and 0.70 for CR for convergent validity ([32]). CFA was conducted using AMOS to validate the hypothesized measurement model (Table 1). Model fit was evaluated using the chi-square test of model fit, comparative fit index (CFI), Tucker–Lewis index (TLI), RMSEA, and standardized root mean square residual (SRMR; [4]). Criteria for an acceptable fit were non-significant chi-square, CFI, and TLI values greater than 0.90, and RMSEA and SRMR less than 0.08 ([33]). Multigroup analysis was then conducted to test for invariance across the Japanese and Chinese samples. Configural invariance was first assessed by specifying the same factor structure across groups without equality constraints ([56]). Metric invariance was tested by constraining factor loadings to be equal across groups, and partial invariance was tested by additionally constraining item intercepts to be equal ([70]). Changes in CFI less than or equal to −0.01 and changes in RMSEA less than or equal to 0.015 were considered indicative of invariance ([43]). After establishing measurement invariance, latent mean differences were explored. The structural model (Appendix D) was then analyzed to assess the relationships hypothesized in the study. The significance of the differences between path coefficients across groups was tested using a multigroup comparison approach, following the recommendations of [11] ([11]).

Multigroup analysis was critical for identifying and understanding the underlying mechanisms of mHealth social acceptance in different cultural contexts. Through this methodical approach, encompassing both descriptive statistics and SEM, we were able to examine not only the hypothesized relationships within our model but also assess their consistency across cultural boundaries.

## 4. Results

Demographic characteristics of the participants in our study revealed distinct patterns across gender, age, education, occupation, smartphone usage, and mHealth service usage (Table 2). In terms of gender, the Japanese cohort showed a slight male predominance (58.40%), whereas the Chinese group had a predominantly female composition (63.80%). The age distribution indicated a younger Japanese cohort, with a significant majority (78.50%) below 20 years of age. By contrast, the Chinese sample was more evenly distributed across age brackets, with a notable proportion (30.3%) falling within the 21–30-year age range. Educational attainment levels differed between the two samples. The Japanese participants had a higher representation at the undergraduate level (48.90%), whereas the Chinese sample had a significant proportion of individuals with master’s degrees or higher (11.00%). Students constituted the majority of both samples, particularly in Japan, where they accounted for 99.60% of respondents, suggesting that our findings may be most representative of the student population. Smartphone usage experience varied, with most participants in both cohorts reporting 3–4 years of experience. Interestingly, a larger segment of the Chinese sample (21.70%) reported an extensive usage history of 8–10 years. Nevertheless, a considerable majority of both Japanese (67.80%) and Chinese (77.20%) participants indicated that they had never used mHealth services.

### 4.1. Measurement Model

#### 4.1.1. Reliability and Validity

To establish the robustness of our mHealth social acceptance measurement model, we evaluated construct reliability and validity for each latent variable (Appendix B). Reliability was assessed using two indicators: Cronbach’s alpha and CR. The convergent validity of each construct was determined using AVE ([67]). In Path A (Japan), prior to adjustment, the construct facilitating conditions (FC) reported a Cronbach’s alpha (0.459) and AVE (0.462), both below the recommended thresholds, suggesting potential issues with internal consistency and convergent validity. This was also reflected in the factor loading for FC1 (0.131), indicating a weak representation of the construct. Adjustments were made to the model to remove FC1 items, an action extended to the China group to maintain consistency. The values then improved, demonstrating better construct reliability and validity (Cronbach’s Alpha for FC increased to 0.720, and AVE to 0.743). Conversely, the China group exhibited satisfactory reliability and validity across all constructs both before and after adjustments, with all CR values well above the threshold of 0.7, and AVEs comfortably exceeding 0.5, suggesting that each construct was measured with a high degree of consistency and accuracy.

#### 4.1.2. CFA

CFA is an advanced statistical technique used to explain the covariance structure of high-dimensional data by identifying a small number of factors ([21]). It entails computing the correlation matrix that the model implicitly implies, which is crucial for determining the model’s parameters and evaluating its validity. The preliminary results of CFA indicated a model that merited refinement to optimize fit (Table 3). The initial chi-square test was significant (χ2 = 2785.125, df = 831, *p* < 0.001), with a chi-square/degrees of freedom (CMIN/DF) ratio of 3.352, suggesting an adequate fit. Although the goodness-of-fit index (GFI) of 0.770 was below the desired threshold of >0.90, it demonstrated potential for improvement. The CFI at 0.902 was close to the recommended value, while the TLI at 0.890 was slightly under the preferable limit. The SRMR value of 0.670 did not exceed the suggested maximum limit of <0.08, and the RMSEA was 0.064, within the acceptable range of <0.08. Following theoretical considerations and empirical diagnostics, we modified the CFA model (Appendix C). These adjustments led to an improved GFI of 0.784, a CFI of 0.909, and a TLI of 0.897, each approaching the recommended standards. The SRMR was reduced to 0.641, and the RMSEA marginally improved to 0.063, further indicating a satisfactory model fit. The post hoc power analysis indicated that the sample size was adequate to detect a close model fit with an RMSEA threshold of 0.05. The calculated power was 1.00, indicating a 100% likelihood of identifying the hypothesized model’s fit ([35]), supporting the robustness of the structural equation model and sample size used in this study.

### 4.2. Measurement Invariance Model

Ensuring that a construct is consistently defined and measured across different groups is essential in comparative research. In assessing measurement invariance, we embarked on a multi-step analysis using a multigroup CFA. The process began with testing for configural invariance to ensure that the factor structure of the model was equivalent across the two groups (Table 4). This step laid the groundwork for further invariance testing ([52]).

The configural invariance model exhibited satisfactory fit indices, with a CMIN/DF (2.901) value below the conventional threshold. The CFI (0.926) and incremental fit index (IFI; 0.927) values exceeding 0.90, and the RMSEA (0.058) within the range were considered to represent an excellent fit for both groups, indicating that the same conceptual framework was applicable across cultures. Upon establishing configural invariance, we proceeded to assess metric invariance, which tests if respondents across groups interpret the constructs in the same way, as indicated by the equality of factor loadings.

However, the metric invariance model (Table 5) revealed significant chi-square differences for both groups, suggesting disparities in the way some items of the constructs were perceived across groups. It indicated significant chi-square differences (CMIN = 63.828, DF = 21, *p* < 0.001), and the normed fit index (NFI) showed minimal change from the configural model (Delta-1 and Delta-2 of 0.003). This suggests that while the factor loadings may vary slightly, the overall fit remains robust; still, significant variance in item interpretation across groups is also indicated. Moreover, the model maintained an adequate fit (CFI = 0.924, IFI = 0.924, RMSEA = 0.058), suggesting the possibility of partial metric invariance. Given these findings, we considered several options including exploring partial invariance, allowing some factor loadings to vary across groups while constraining others ([62]), thereby providing a nuanced understanding of which specific items contribute to the lack of full metric invariance. Allowing for partial metric invariance resulted in a better fit compared to the full metric model, with an improved CMIN/DF of 2.866, GFI of 0.800, and unchanged CFI, IFI, and RMSEA values. The chi-square statistic was not significant (CMIN = 26.355, DF = 18, *p* = 0.092), with fit indices within acceptable ranges (CFI = 0.926, IFI = 0.926), suggesting that relaxing some of the factor loading constraints did not deteriorate the model fit significantly.

Scalar invariance, necessary for latent mean comparisons ([51]), was evidenced by the strongest model fit indices (CMIN/DF = 2.745, CFI = 0.93, IFI = 0.931, RMSEA = 0.055). The chi-square statistics remained significant (CMIN = 28.593, DF = 18, *p* = 0.054); however, the changes in NFI were still marginal (Delta-1 of 0.001, Delta-2 of 0.002), indicating that the model’s constraints are well tolerated, confirming the model’s robustness, and validating the cross-cultural comparability of latent variables in the context of mHealth acceptance. Our sequential approach from configural to full metric, partial metric, and scalar invariance provided a comprehensive picture of the model’s applicability across cultural boundaries. Although full metric invariance was not demonstrated, the establishment of partial and scalar invariance indicated that the latent constructs could be compared meaningfully across groups. This step was imperative for subsequent comparisons of latent means and theoretical interpretations.

### 4.3. Latent Mean Comparison

Latent mean comparisons in multigroup analysis explain the differences across groups ([22]). It helps researchers determine whether there are actual variations in the average levels of constructs between groups ([83]). The latent mean (Table 6) for FC showed no significant difference between the two countries (estimate = −0.02, SE = 0.178, CR = −0.115, *p* = 0.908 for both groups), indicating that the contextual factors enabling or hindering behaviors or actions are perceived similarly in Japan and China. Similarly, the influence of societal norms and relationships, captured under the Social Influence (SI) construct, did not differ significantly between the groups (estimate = −0.322, SE = 0.396, CR = −0.813, *p* = 0.416 for both groups). These findings suggest that both FC and SI operate with a comparable level of impact in the two distinct cultural contexts, providing a robust basis for the interpretation of these constructs as culturally invariant in our model. This supports the theoretical assumption that these latent variables have equivalent meanings and salience across societies.

### 4.4. Comparative Path Analysis

Comparative path analysis was employed to dissect the structural interplay among several theoretical constructs assessed in the participants from Japan and China (Figure 1) Comparative path analysis in multigroup settings involves testing hypothesized causal relationships between variables across different groups ([25]). This is in tandem with the hypotheses of the study. The relationship between FC and effort expectancy (EE) was found to be strong in both groups, but notably more pronounced in China (Japan: β = 0.56, SE = 0.079, CR = 7.069, *p* < 0.001; China: β = 0.904, SE = 0.044, CR = 20.637, *p* < 0.001).

Performance expectancy (PE), derived from SI, was significant in both Japanese (β = 0.328, SE = 0.048, CR = 6.807, *p* < 0.001) and Chinese cohorts (β = 0.733, SE = 0.038, CR = 19.336, *p* < 0.001), with the impact being significantly stronger in China.

Trust (TR), influenced by both health consciousness (HC) and SI, exhibited a substantial effect in both Japan and China, with the influence of SI on TR appearing stronger in China (Japan: TR ← HC: β = 0.285, SE = 0.062, CR = 4.61, *p* < 0.001; TR ← SI: β = 0.248, SE = 0.044, CR = 5.627, *p* < 0.001; China: TR ← HC: β = 0.385, SE = 0.049, CR = 7.869, *p* < 0.001; TR ← SI: β = 0.561, SE = 0.043, CR = 13.019, *p* < 0.001).

Behavioral intentions (BIs) to use mHealth technologies were significantly predicted by EE and PE in Japan (EE: β = 0.34, SE = 0.094, CR = 3.61, *p* < 0.001; PE: β = 0.441, SE = 0.123, CR = 3.58, *p* < 0.001), whereas in China, TR had the most significant effect (β = 0.674, SE = 0.063, CR = 10.745, *p* < 0.001), though PE also contributed significantly to BI (β = 0.378, SE = 0.065, CR = 5.765, *p* < 0.001). Surprisingly, PR did not show a significant effect on BI in either Japan or China (Japan: β = 0.000, SE = 0.053, CR = 0.009, *p* = 0.993; China: β = −0.004, SE = 0.027, CR = −0.136, *p* = 0.892), suggesting that within these cultures, risk perception may not play a pivotal role in the acceptance of mHealth (Table 7).

## 5. Discussion

The substantial role in facilitating mHealth technology acceptance has been well-documented within the UTAUT model and its extension; nevertheless, our results emphasize the variable weight of these constructs across cultural boundaries. The post hoc power of 1.00 indicates strong support for the findings, complementing the model fit indices and suggesting that the sample size was adequate to detect the hypothesized relationships within the model. This additional validation enhances confidence in the robustness of the cross-cultural comparisons. While Japanese youth recognize the role of facilitating conditions, the criticality of efficient and supportive infrastructure is paramount for their Chinese counterparts. The adoption of mHealth solutions among the youth populations is subject to a spectrum of influential factors including performance expectancy, trust, effort expectancy, and health consciousness ([10]). Furthermore, the influence of cultural and regional disparities underscores the imperative for customized strategies aimed at fostering mHealth acceptance across diverse contexts ([27]).

This comparative analysis not only highlights the unique cultural nuances influencing technology acceptance in these countries but also aligns these trends with global movements in mHealth practices. For example, while Japanese youth exhibit a cautious yet growing acceptance of mHealth, similar patterns are evident in Western contexts, where there is increasing trust in digital health solutions amid privacy concerns ([30]). Conversely, the enthusiastic uptake among Chinese youth mirrors the rapid digital transformation in emerging economies ([3]; [31]; [72]), characterized by a leapfrogging over traditional healthcare solutions. Despite relatively low usage rates of mHealth applications among Vietnamese youth ([77]), there is notable openness and satisfaction among those who do utilize such services, suggesting a potential untapped market for mHealth innovations that could apply to both Japanese and Chinese youth. Furthermore, the importance of mHealth in managing chronic conditions such as diabetes reveals its potential to significantly enhance health outcomes ([88]). These tools indicate substantial potential for facilitating better health outcomes. However, sustaining user engagement over time may pose a challenge for mHealth initiatives in Japan and China, which must address this issue through tailored, culturally appropriate interventions that resonate with local user preferences and health management behaviors. This suggests that for broader adoption, mHealth applications must not only be accessible but also deeply integrated with elements that engage and retain a diverse user base. Recent innovations in mHealth have demonstrated considerable promise in addressing stigmatization and enhancing health outcomes among youth, particularly in the context of prevention and care for manageable diseases ([54]), noting its potential to provide culturally appropriate and tailored interventions.

The receptiveness and preference for health-related smartphone applications among youth indicate not only a high acceptance but also a significant perceived utility, reflecting their openness to leveraging technology for managing health-related issues and their recognition of the benefits these applications provide in terms of accessibility, convenience, and personalized health interventions ([74]). These studies suggest that mHealth platforms can significantly impact youth by offering accessible, stigma-reducing health interventions that are adaptable to their diverse needs, which could be similarly effective in the cultural contexts of Japan and China, where mobile technology usage among youth is prevalent ([54]). The nascent yet rapidly expanding application of mHealth for mental health in China provides significant benefits in accessibility and patient outcomes, despite challenges such as data privacy and user acceptability ([37]). Research further highlights the potential of mHealth to enhance healthcare efficiency, noting that perceived usefulness and ease of use are critical predictors of adoption intentions among users ([23]). Moreover, studies have found that social influence and perceived ease of use significantly influence the intention to use mHealth among older adults, a factor likely extrapolatable to younger demographics given their higher propensity for technology adoption and susceptibility to peer influence ([13]). Despite the high risk of bias in existing studies, the integration of culturally adapted mHealth solutions has demonstrated improvements in mental health outcomes, emphasizing the importance of tailored mHealth strategies to meet specific healthcare needs in diverse regional contexts ([37]). These insights are crucial for developing effective mHealth platforms that resonate with the dynamic and culturally diverse youth populations in Japan and China.

This study’s path analysis provided a nuanced lens through which to view the factors influencing the acceptance of mHealth technologies among the youth in Japan and China, revealing distinct patterns that both conform to and diverge from the existing literature. Contrary to the assumptions laid out by [15] ([15]) and [76] ([76]), perceived risk was not a significant deterrent to mHealth adoption for either Japanese or Chinese youth (Japan: β = 0.000, SE = 0.053, CR = 0.009, *p* = 0.993; China: β = −0.004, SE = 0.027, CR = −0.136, *p* = 0.892). This finding suggests a potential paradigm shift in the perception of digital health risks among younger populations, possibly indicative of greater familiarity and comfort with technology that transcends apprehension about possible adverse outcomes. Privacy and regulatory assurances play a crucial role in shaping mHealth adoption, particularly in regions with differing levels of institutional trust. In China, stronger governmental oversight may mitigate privacy concerns, whereas, in Japan, users’ hesitancy could be linked to higher sensitivity toward personal data security. Addressing these concerns through transparent data policies and user control features can enhance trust and facilitate adoption. Trust was confirmed as a fundamental factor in attracting and retaining mHealth users, significantly influencing behavioral intentions across both cultural contexts. The trust differences observed between Japan and China may be influenced by Hofstede’s individualism–collectivism dimensions ([20]; [85]). Japan’s high individualism suggests a preference for personal control and trustworthiness ([12]), whereas China’s collectivist culture emphasizes group consensus and institutional trust ([90]). Beyond trust, the individualism-collectivism dimension also influences other key mHealth adoption factors. In collectivist societies, social influence plays a stronger role ([55]), as individuals rely on group norms and institutional endorsements when adopting new technologies. In contrast, in individualistic societies, perceived usefulness is more influential ([34]), as users independently assess the benefits of technologies. Additionally, risk perception in collectivist settings is mitigated by institutional trust, whereas individualistic societies emphasize privacy and data security concerns ([73]). This aligns with Rogers’ ([80]) diffusion of innovations theory ([91]), which posits that trust is instrumental in the rate of technology adoption. The significance of performance expectancy in Japan (β = 0.441, SE = 0.123, CR = 3.58, *p* < 0.001) underscores the value that Japanese youth place on the practical benefits of mHealth, whereas the stronger influence of effort expectancy in China (β = 0.904, SE = 0.044, CR = 20.637, *p* < 0.001) suggests a prioritization of usability and ease of integration into daily life. These findings reveal significant cultural differences in mHealth adoption. In Japan, the emphasis on performance expectancy suggests that mHealth applications should prioritize efficiency and functional benefits. In China, where trust plays a stronger role, institutional credibility and regulatory assurances are crucial for adoption. These insights highlight the need for culturally tailored mHealth strategies that align with local expectations and behavioral tendencies. This difference in emphasis echoes Goodman et al.’s (2023) findings, highlighting that while the utility and functionality of mHealth are critical for adoption, cultural contexts may determine which aspect is more influential. Health consciousness uniformly influenced the perception of mHealth in both cohorts positively, suggesting that the more individuals value their health, the more likely they are to accept mHealth solutions. This personal characteristic, alongside the environmental factor of social influence, which was particularly salient in China (β = 0.561, SE = 0.043, CR = 13.019, *p* < 0.001), demonstrates the importance of aligning mHealth initiatives with individuals’ intrinsic values and the prevailing social ethos. Notably, facilitating conditions—defined as the degree to which individuals believe that the existing infrastructure supports mHealth use—were found to be a significant factor, especially in China. This aligns with the values reported, indicating a strong correlation between facilitating conditions and the perceived ease of mHealth technology use in China (β = 0.904, SE = 0.044, CR = 20.637, *p* < 0.001). This effect size reduces the influence of facilitating conditions in Japan (β = 0.56, SE = 0.079, CR = 7.069, *p* < 0.001), suggesting a greater emphasis on accessible healthcare infrastructure as a determinant of mHealth adoption among Chinese youth.

### 5.1. Recommendations

The research underscores the need for culturally and contextually adapted mHealth solutions that cater to the distinct characteristics of youth populations in Japan and China. This approach is crucial not only for youth in general but also for addressing the specific needs of physically challenged youths. To this end, privacy assurance features such as voice recognition, adjustable text sizes, and screen reader compatibility will make these applications more usable for youths with physical challenges. Furthermore, the findings highlight the importance of incorporating robust educational components within these technologies. Educating young adults about the functionalities and benefits of mHealth through interactive tutorials and real-time assistance can enhance user engagement and proficiency. For physically challenged youth, specialized training modules should be developed to provide additional support and foster independence in managing their health conditions.

Community integration features within mHealth platforms can significantly enhance user experience and support. Social forums, peer support groups, and mentorship programs within the apps can facilitate community building, offering valuable opportunities for youth to share experiences and support. To enhance accessibility and cultural relevance, mHealth applications should integrate localized language support and customized content addressing unique healthcare challenges present in Japan and China. Enhanced security features and transparent data policies can address privacy concerns in Japan, boosting adoption. AI-driven chatbots in Japanese and Mandarin can further engage users with instant health guidance. Additionally, leveraging WeChat-based health services and government-endorsed platforms may drive wider adoption in China. These strategies can strengthen user trust, usability, and long-term engagement. Additionally, ongoing research and development driven by feedback from these populations will enable continuous improvement of mHealth solutions. Incorporating feedback mechanisms within the apps to collect insights on user experience can guide future enhancements, making these tools more responsive to the needs of young users, regardless of their health condition. It is also important for developers and policymakers to create targeted strategies that enhance user engagement. In Japan, where there is hesitation towards mHealth, strategies should focus on raising public awareness about the benefits and security measures of these technologies. Informational campaigns and demonstrations could communicate the value and safety of mHealth solutions, aiming to alleviate concerns. In China, where there is already a positive perception towards these technologies, efforts should be directed towards optimizing usability to ensure seamless integration of mHealth tools into daily routines. Enhancing interface design and functionality can capitalize on positive attitudes, promoting wider adoption among the youth. These strategies should be culturally aligned and leverage the widespread use of mobile technology among young people in both regions.

Implementing accessible, stigma-free health interventions through mHealth platforms can meet the diverse needs of young populations and potentially transform public health landscapes. By tailoring strategies to align with the unique cultural and technological environments of Japan and China, mHealth initiatives have the potential to enhance their effectiveness, thereby making a substantial impact within these communities and potentially serving as a blueprint for similar strategies on a global scale.

### 5.2. Limitations and Future Research

Although this study provides valuable insights, it has certain limitations that suggest directions for future research. The study’s focus on students may limit the generalizability of the findings to non-university youth and older populations, who may have different technology acceptance patterns. Additionally, the cross-sectional nature of the study does not allow for the exploration of changes over time in attitudes toward mHealth. Future research should consider longitudinal studies to track changes in perceptions and usage of mHealth over time, as well as expand the demographic scope to include non-university youth and older age groups. While this research examines the initial adoption of mHealth solutions, it is essential to also consider their long-term impact, which may facilitate broader adoption and encourage behavioral changes that promote positive health outcomes. Sustained user engagement is critical to ensuring the continued effectiveness of these interventions. Future research should explore strategies such as personalized and user-friendly mechanisms to maintain user interest and enhance health outcomes over time. Further, exploring the impact of emerging technologies such as artificial intelligence and machine learning on mHealth acceptance could provide deeper insights into the evolving landscape and future of health technology. In conclusion, this study underscores the importance of considering cultural and regional differences in the development and implementation of mHealth technologies. Although the post hoc power analysis indicated that the sample size was adequate, its retrospective nature and reliance on observed parameters are acknowledged as limitations. Future studies could enhance their methodology by incorporating prospective power calculations during the design phase or by utilizing alternative approaches to obtain more robust estimations. By aligning mHealth initiatives with the intrinsic values and technological expectations of target populations, developers and policymakers can greatly enhance the societal acceptance and effectiveness of these health interventions.

## 6. Conclusions

Empirical investigations into mHealth acceptance have revealed diverse patterns and determinants across cultural landscapes, with the variability in mHealth adoption among youth being influenced by demographic characteristics, health literacy levels, and technological familiarity. The high post hoc power (1.00) confirms the adequacy of the sample size in supporting the hypothesized model, confirming the robustness of the statistical analyses, and ensuring confidence in the reliability of the results and their relevance to the cultural and contextual determinants of mHealth adoption. The findings advocate for culturally sensitive mHealth interventions that resonate with the specific values, communication styles, and health engagement behaviors of target populations. Moreover, cross-cultural research in mHealth acceptance highlights the necessity of tailoring solutions to align with localized health practices, language preferences, and technology usage norms. This study illuminates the critical role of cultural adaptation in enhancing mHealth’s effectiveness and user engagement, advocating for mHealth platforms that reflect an understanding of and respect for cultural idiosyncrasies in health perceptions and technology interactions. Additionally, the methodological approach, anchored by the extended UTAUT, proved instrumental in dissecting these complex cultural dynamics and allowed for a nuanced analysis that not only highlighted the differential impacts of specific constructs but also provided a robust mechanism for comparing these effects across diverse settings. Our study contributes to the broader discourse on mHealth by illustrating how localized cultural factors intertwine with global technology adoption trends, offering insights that are vital for policymakers and developers aiming to foster mHealth acceptance on a worldwide scale. These insights are crucial for developing effective mHealth platforms that resonate with the dynamic and culturally diverse youth populations in Japan and China.

## Figures and Tables

**Figure 1 behavsci-15-00213-f001:**
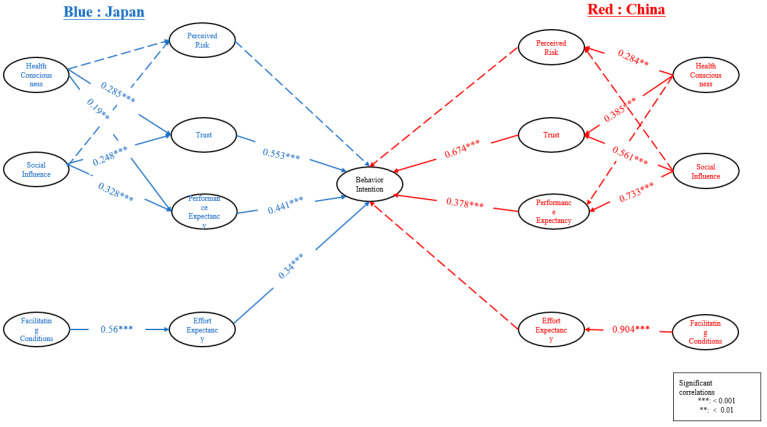
Path model of mHealth acceptance among youth in Japan and China. The figure illustrates the structural relationships influencing mHealth acceptance in Japan (blue paths) and China (red paths). Path coefficients (β) and significance levels (*p* < 0.05, *p* < 0.01, *p* < 0.001) indicate the strength and statistical significance of the relationships. Solid arrows indicate significant paths, and dashed arrows indicate non-significant paths. EE: effort expectancy; FC: facilitating conditions; HC: health consciousness; TR: trust; SI: social influence; PE: performance expectancy; BI: behavioral intention; PR: perceived risk.

**Table 1 behavsci-15-00213-t001:** Framework components.

Dimensions	Constructs	Content
Psychological dimensions	Perceived Risk (PR)	Perceived risk is defined as the extent to which people judge the possible adverse outcomes of mHealth ([6]; [50]).
	Trust (TR)	Trust is defined as the level of confidence that users have in the reliability and integrity of mHealth technologies, and is an important factor in attracting and retaining users ([8]; [61]).
	Performance Expectancy (PE)	Performance expectancy is defined as the degree to which a person perceives that a new technology or service is useful for improving job performance ([7]; [68]).
	Effort Expectancy (EE)	Effort expectancy is defined as the subjective perception of the difficulty of a technique ([5]; [66]).
Personal characteristics	Health Consciousness (HC)	Health consciousness is defined as the degree to which individuals value their health ([19]; [65]).
Environmental characteristics	Social Influence (SI)	Social influence is defined as the influence of the perceptions of others on the willingness to use mHealth ([60]; [69]).
Conditions of use	Facilitating Conditions (FC)	Facilitation conditions are the extent to which individuals perceive that the existing infrastructure can support mHealth use ([47]; [89]).

**Table 2 behavsci-15-00213-t002:** Demography characteristics.

Distributions		Japan		China		Total	
		N	%	N	%	N	%
	Male	136	58.40%	122	36.20%	258	45.30%
Gender	Female	97	41.60%	215	63.80%	312	54.70%
Total		233	100.00%	337	100.00%	570	100.00%
Age	≤20	183	78.50%	212	62.90%	395	69.30%
	21–30	50	21.50%	102	30.30%	152	26.67%
	31–40	0	0.00%	20	5.90%	20	3.51%
	41–50	0	0.00%	2	0.60%	2	0.35%
	51–60	0	0.00%	0	0.00%	1	0.18%
	≥60	0	0.00%	1	0.30%	0	0.00%
Total		233	100.00%	337	100.00%	570	100.00%
Education level	Junior School	0	0.00%	7	2.10%	7	1.20%
	High School	108	46.40%	169	50.10%	277	48.60%
	Vocational College	8	3.40%	0	0.00%	8	1.40%
	Undergraduate Degree	114	48.90%	124	36.80%	238	41.80%
	Master’s Degree	3	1.30%	27	8.00%	30	5.30%
	Doctorate Degree and Higher	0	0.00%	10	3.00%	10	1.80%
Total		233	100.00%	337	100.00%	570	100.00%
Occupation	Students	232	99.60%	312	92.60%	544	95.40%
	Enterprises (Foreign or Private Owned)	1	0.40%	9	2.70%	10	1.80%
	Medical Practitioners	0	0.00%	7	2.10%	7	1.20%
	Freelancers	0	0.00%	5	1.50%	5	0.90%
	Education Sector	0	0.00%	3	0.90%	3	0.50%
	Public Servants	0	0.00%	1	0.30%	1	0.20%
Total		233	100.00%	337	100.00%	570	100.00%
Smartphone usage experience (Year)	1–3 years	10	4.30%	23	6.80%	33	5.80%
	4–7 years	142	60.90%	144	42.70%	286	50.20%
	8–10 years	72	30.90%	97	28.80%	169	29.60%
	More than 10 years	9	3.90%	73	21.70%	82	14.40%
Total		233	100.00%	337	100.00%	570	100.00%
mHealth service (app Usage Experience)	I Have Never Used It	158	67.80%	260	77.20%	418	73.30%
	I Have Used It	75	32.20%	77	22.80%	152	26.70%
Total		233	100.00%	337	100.00%	570	100.00%

**Table 3 behavsci-15-00213-t003:** Confirmatory factor analysis (CFA).

Indices	Recommended Value	Initial Value	Final Value
P	Insignificant	0.000	0.000
CMIN/DF	3–5	3.352	3.29
GFI	>0.90	0.770	0.784
CFI	>0.90	0.902	0.909
TLI	>0.90	0.890	0.897
SRMR	<0.08	0.670	0.641
RMSEA	<0.08	0.064	0.063

Note: CMIN/DF: chi-square/degrees of freedom ratio; GFI: goodness-of-fit index; CFI: comparative fit index; TLI: Tucker–Lewis index; SRMR: standardized root mean square residual; RMSEA: root mean square error of approximation.

**Table 4 behavsci-15-00213-t004:** Goodness-of-fit statistics for the multigroup confirmatory factor analysis.

	Model Fit					
	*p*-Value	CMIN/DF	GFI	CFI	IFI	RMSEA
Configural invariance	<0.001	2.901	0.803	0.926	0.927	0.058
Full Metric invariance	<0.001	2.905	0.797	0.924	0.924	0.058
Partial Metric invariance	<0.001	2.866	0.8	0.926	0.926	0.057
Scalar invariance	<0.001	2.745	-	0.93	0.931	0.055

Note: CMIN/DF: chi-square/degrees of freedom ratio; GFI: goodness-of-fit index; CFI: comparative fit index; IFI: incremental fit index; RMSEA: root mean square error of approximation.

**Table 5 behavsci-15-00213-t005:** Invariance measurement result.

Model	DF	CMIN	*p*	NFI	IFI	RFI	TLI
				Delta-1	Delta-2	rho-1	rho2
Full Metric	21	63.828	0	0.003	0.003	0	0
Partial Metric	18	26.355	0.092	0.001	0.001	−0.001	−0.002
Scalar Invariance	18	28.593	0.054	0.001	0.002	−0.001	−0.001

DF: degrees of freedom; CMIN: chi-square minimum; NFI: normed factor index; IFI: incremental factor index; RFI: relative fit index; TLI: Tucker–Lewis index.

**Table 6 behavsci-15-00213-t006:** Latent mean comparison.

	Japan				China			
	Estimate	SE	CR	*p*	Estimate	SE	CR	*p*
**FC**	−0.02	0.178	−0.115	0.908	−0.02	0.178	−0.115	0.908
**HC**	−2.076	0.867	−2.395	0.017	−2.076	0.867	−2.395	0.017
**SI**	−0.322	0.396	−0.813	0.416	−0.322	0.396	−0.813	0.416

Note: SE: standard errors; CR: critical ratios; FC: facilitating conditions; HC: health consciousness; SI: social influence.

**Table 7 behavsci-15-00213-t007:** Comparative path analysis of the constructs.

	Japan				China			
	Estimate	SE	CR	*p*	Estimate	SE	CR	*p*
EE<--FC	0.56	0.079	7.069	***	0.904	0.044	20.637	***
PR<--HC	0.153	0.13	1.184	0.236	0.284	0.102	2.785	0.005
TR<--HC	0.285	0.062	4.61	***	0.385	0.049	7.869	***
TR<--SI	0.248	0.044	5.627	***	0.561	0.043	13.019	***
PR<--SI	0.106	0.09	1.182	0.237	0.203	0.084	2.418	0.016
PE<--HC	0.19	0.069	2.766	0.006	0.045	0.037	1.193	0.233
PE<--SI	0.328	0.048	6.807	***	0.733	0.038	19.336	***
BI<--EE	0.34	0.094	3.61	***	0.055	0.055	1.005	0.315
BI<--PE	0.441	0.123	3.58	***	0.378	0.065	5.765	***
BI<--TR	0.553	0.14	3.945	***	0.674	0.063	10.745	***
BI<--PR	0.000	0.053	0.009	0.993	−0.004	0.027	−0.136	0.892

Note: *** *p*-value < 0.01; SE: standard error; CR: critical ratio; EE: effort expectancy; FC: facilitating conditions; HC: health consciousness; TR: trust; SI: social influence; PE: performance expectancy; BI: behavioral intention; PR: perceived risk.

## Data Availability

The corresponding author will provide access to the data upon request.

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
