# Peer review of "Cross-Analysis of mHealth Social Acceptance Among Youth: A Comparative Study Between Japan and China"

_behavsci, 2025, doi:10.3390/bs15020213_

Round 1
Reviewer 1 Report
Comments and Suggestions for Authors
1. Introduction: I understand there are not an abundance of research comparing both, but are these references (Cao et al, 2022; To et al., 2019) the only references on acceptance of mHealth in Japan and China? Otherwise, well written and interesting.
2. Theoretical Framework and Proposed Model
Some of the wording for your hypotheses are not clear to readers. For example, H5a to H5c, are you testing an interaction or moderation effect? Health consciousness is your X and mHealth perceptions is your outcome. But are PR, TR and perceived usefulness mediators? Again your hypotheses in H6 sound like they could be both mediators and moderators, but you use the word "modulate" which sounds like interaction to me. What is the outcome and which are the mediator/interaction? Please make your hypotheses very very clear (use words like mediator/mediate and interaction/moderation).
3. Materials and Methods
Line 170-172: I would suggest using simple words when it is possible. For example: "We used standardized questionnaires to assess young people's perceptions, usage intentions, and potential future use of mHealth technologies."
Lines 182, 189: Is a single response from a single user? If so, please use for example: "...gathered responses from 337 students."
How much overlap was there between the Japanese vs. Chinese samples in the questions asked? How was the data merged?
Line 197: "to summarize the sample characteristics shown in Table 2." Again, avoid words like "central tendencies" when they are not needed (ex: you can just say mean or median)
Author Response
Comment 1: Introduction
I understand there are not an abundance of research comparing both, but are these references (Cao et al, 2022; To et al., 2019) the only references on acceptance of mHealth in Japan and China? Otherwise, well written and interesting.
Response 1: Thank you for your comment. We acknowledge the importance of a comprehensive literature review on mHealth acceptance among youth in Japan and China. However, despite an extensive search, we found that empirical studies specifically examining mHealth adoption among youth in both Japan and China remain scarce. The references cited (Cao et al., 2022; To et al., 2019) are among the few available studies that focus on this demographic in these two countries.
Comment 2: Theoretical Framework and Proposed Model
Some of the wording for your hypotheses are not clear to readers. For example, H5a to H5c, are you testing an interaction or moderation effect? Health consciousness is your X and mHealth perceptions is your outcome. But are PR, TR and perceived usefulness mediators? Again your hypotheses in H6 sound like they could be both mediators and moderators, but you use the word "modulate" which sounds like interaction to me. What is the outcome, and which are the mediator/interaction? Please make your hypotheses very clear (use words like mediator/mediate and interaction/moderation).
Response 2: We have revised the hypotheses to explicitly differentiate between mediation and moderation effects. H5a–H5c now clearly state that perceived risk, trust, and perceived usefulness mediate the relationship between health consciousness and behavioral intention. H6 has been revised to explicitly state that social influence moderates the relationships between health consciousness and perceived risk, trust, and performance expectancy. These revisions are reflected in the Theoretical Framework and Proposed model section.
Comment 3: Materials and Methods
Line 170-172: I would suggest using simple words when it is possible. For example: "We used standardized questionnaires to assess young people's perceptions, usage intentions, and potential future use of mHealth technologies."
Lines 182, 189: Is a single response from a single user? If so, please use for example: "...gathered responses from 337 students."
How much overlap was there between the Japanese vs. Chinese samples in the questions asked? How was the data merged?
Line 197: "to summarize the sample characteristics shown in Table 2." Again, avoid words like "central tendencies" when they are not needed (ex: you can just say mean or median)
Response 3: We have simplified the language as you suggested throughout the Materials and Methods section for improved readability.
Regarding overlap and data merging: The survey questions were identical for both samples, ensuring consistency in the constructs measured. Data from Japan and China were not merged but analysed separately using multi-group Structural Equation Modelling (SEM) to preserve the distinctiveness of each dataset while enabling structured cross-group comparisons.
Reviewer 2 Report
Comments and Suggestions for Authors
Strengths of the Paper
Timely Topic:
The paper addresses a timely and relevant issue, exploring the cross-cultural acceptance of mHealth technologies among youth in Japan and China. This is an increasingly growing area of interest due to the rise in mobile technologies, smartphones, and mHealth in general.
Methodological Rigor:
The extended UTAUT framework and the use of Confirmatory Factor Analysis and Structural Equation Modelling to validate the constructs adds methodological rigour to the paper.
Key Areas for Improvement:
1. Survey Instrument Details
- The paper states that standardized questionnaires were administered to assess young people's perceptions and usage intentions regarding mHealth technologies. However, the survey instrument itself, or a link to its full content, is not provided in the manuscript or appendices. This is a problem and its absence does not support the scientific process (repeatability). Including the actual questionnaire or detailed items is very important for transparency, replicability, and validating the study’s findings. Please include the full questionnaire in an appendix or directly in the body of the paper or providing a link to it where it can be easily accessed. This is a major fault and needs to be corrected. It would also be good to include references to other studies that have used the same instrument, thereby increasing the credibility of using this particular survey.
2. Sampling Limitations
- The study uses a harmonized sampling methodology, but the reliance on university students as participants limits the generalizability of the findings. This limitation should be acknowledged more explicitly in the manuscript.
3. Cultural Analysis Depth
- The cultural analysis focuses on individualism versus collectivism but does not delve deeply into how these cultural dimensions shape the specific factors influencing mHealth adoption. Expanding this section with more cultural context would strengthen the paper’s contribution.
4. Statistical Reporting
- The statistical analyses appears to be thorough, but the paper could benefit from a clearer explanation of the differences between Japan and China (ie., path coefficients). Their practical implications could also be discussed in greater detail.
5. Privacy and Ethical Considerations
- The paper mentions privacy and ethical issues as factors in mHealth adoption, however, there is little elaboration on how these were addressed in the study. (e.g., how they might impact adoption in the two countries). Expanding this would enhance the discussion section of the paper.
6. Recommendations Section (section 5.2)
- The recommendations are helpful but could be more actionable. e.g., instead of stating that mHealth applications should be more accessible, if the authors could provide specific examples, (Eg.., incorporating local languages or addressing unique cultural barriers, etc.), this would strengthen the paper.
7. Figures and Visuals
- Some figures (e.g., Path Model (Figure 1)), could be clearer by adding annotations or explanations. This would help readers unfamiliar with CFA or SEM to more easily understand. Also, including legends or descriptions would improve ease of understanding.
8. Clarity and Language
* The manuscript is well-written overall, but there are areas where clarity can be improved (e.g, explanation of the theoretical framework).
9. Reference Alignment
- Ensure all references cited in the text are fully aligned with the reference list. A few citations appear incomplete or may require further detail.
-
10. Suggestions for Future Research (considerations for section 5.3)
-
- Longitudinal Studies:
- Future research could explore changes in mHealth acceptance over time, especially since digital health technologies continue to evolve.
- Broader Demographics:
- Noting that by expanding the participant pool would promote better generalizability of the work conduction. e.g., include non-university youth and/or older age groups.
- Emerging Technologies:
- The study could explore the role of AI-driven mHealth apps or other new opportunities in this area. It may be relevant as it may influence acceptance in culturally distinct populations.
- Longitudinal Studies:
Author Response
Comment 1: Survey Instrument Details
The paper states that standardized questionnaires were administered to assess young people's perceptions and usage intentions regarding mHealth technologies. However, the survey instrument itself, or a link to its full content, is not provided in the manuscript or appendices. This is a problem and its absence does not support the scientific process (repeatability). Including the actual questionnaire or detailed items is very important for transparency, replicability, and validating the study’s findings. Please include the full questionnaire in an appendix or directly in the body of the paper or providing a link to it where it can be easily accessed. This is a major fault and needs to be corrected. It would also be good to include references to other studies that have used the same instrument, thereby increasing the credibility of using this particular survey.
Response 1: Thank you for this insightful recommendation. We acknowledge this important recommendation. The survey instrument has been included as Appendix E, providing the full survey items and references to validated studies.
Comment 2: Sampling Limitations
The study uses a harmonized sampling methodology, but the reliance on university students as participants limits the generalizability of the findings. This limitation should be acknowledged more explicitly in the manuscript.
Response 2: We have acknowledged the limitation regarding the reliance on university students and its potential impact on generalizability in the Limitations and Future Research section of the manuscript. This section also discusses the need for broader demographic inclusion in future research."
Comment 3: Cultural Analysis Depth
The cultural analysis focuses on individualism versus collectivism but does not delve deeply into how these cultural dimensions shape the specific factors influencing mHealth adoption. Expanding this section with more cultural context would strengthen the paper’s contribution.
Response 3: We have expanded the discussion on how cultural dimensions shape the specific factors influencing mHealth adoption. This revision has been incorporated into the discussion section of the manuscript.
Comment 4: Statistical Reporting
The statistical analyses appears to be thorough, but the paper could benefit from a clearer explanation of the differences between Japan and China (ie., path coefficients). Their practical implications could also be discussed in greater detail.
Response 4: We appreciate this suggestion. The interpretation of path coefficient differences between Japan and China has been clarified, and their practical implications have been expanded upon in the Discussion section. of the manuscript.
Comment 5: Privacy and Ethical Considerations
The paper mentions privacy and ethical issues as factors in mHealth adoption, however, there is little elaboration on how these were addressed in the study. (e.g., how they might impact adoption in the two countries). Expanding this would enhance the discussion section of the paper.
Response 5: We have expanded the discussion on the importance of privacy and ethical considerations in mHealth adoption. We have expanded the discussion section to elaborate on how these concerns were addressed in our study and their implications for adoption in Japan and China
Comment 6: Recommendations Section (section 5.2)
The recommendations are helpful but could be more actionable. e.g., instead of stating that mHealth applications should be more accessible, if the authors could provide specific examples, (E.g., incorporating local languages or addressing unique cultural barriers, etc.), this would strengthen the paper.
Response 6: We appreciate this feedback. The recommendations section has been revised to provide more actionable insights. Specifically, we now include examples such as integrating localized language support and developing culturally tailored content to address healthcare challenges unique to Japan and China
Comment 7: Figures and Visuals
Some figures (e.g., Path Model (Figure 1)), could be clearer by adding annotations or explanations. This would help readers unfamiliar with CFA or SEM to more easily understand. Also, including legends or descriptions would improve ease of understanding.
Response 7: We appreciate these helpful suggestions. We have revised Figure 1 (Path Model) to improve clarity by adding annotations, a legend, and an expanded figure description. The annotations now differentiate relationships in Japan (blue paths) and China (red paths), while also specifying significant (solid arrows) and non-significant (dashed arrows) paths. The updated legend defines key model constructs and statistical notations, ensuring accessibility for readers unfamiliar with CFA and SEM. These changes have been incorporated into the manuscript to enhance clarity and understanding
Comment 8: Clarity and Language
* The manuscript is well-written overall, but there are areas where clarity can be improved (e.g, explanation of the theoretical framework).
Response 8: We have improved the clarity of the theoretical framework by refining the definitions of key constructs and ensuring a clearer articulation of their theoretical relationships. Additionally, we have explicitly differentiated between mediation and moderation effects, making the conceptual model more transparent and readable
Comment 9: Reference Alignment
Ensure all references cited in the text are fully aligned with the reference list. A few citations appear incomplete or may require further detail.
Response 9: We have carefully reviewed the reference list to ensure full alignment with in-text citations. All citations have been cross-checked, and missing details have been added where necessary.
Comment 10: Suggestions for Future Research (considerations for section 5.3)
Longitudinal Studies: Future research could explore changes in mHealth acceptance over time, especially since digital health technologies continue to evolve.
Broader Demographics: Noting that by expanding the participant pool would promote better generalizability of the work conduction. e.g., include non-university youth and/or older age groups.
Emerging Technologies: The study could explore the role of AI-driven mHealth apps or other new opportunities in this area. It may be relevant as it may influence acceptance in culturally distinct populations.
Response 10: We appreciate your recommendation to expand the discussion on future research directions. We have ensured that our Limitations and Future Research section explicitly addresses the need for longitudinal studies, expanding research to non-university youth and older populations, and exploring emerging technologies
Reviewer 3 Report
Comments and Suggestions for Authors
This paper presents an empirical cross-cultural analysis of mobile health (mHealth) acceptance among youth in Japan and China, utilizing an extended version of the Unified Theory of Acceptance and Use of Technology (UTAUT) framework. It provides valuable insights into how cultural factors influence mHealth adoption, behavioral intentions, and technology acceptance. The following comments are provided for consideration: [1] The interpretation of cultural differences overly relies on Hofstede’s Individualism-Collectivism framework and overlooks other important cultural and policy-related factors. It would be beneficial to explore Japan’s lower mHealth adoption rates beyond individualism, including aspects such as trust in the medical system or strict privacy regulations. Similarly, for China’s higher mHealth adoption, factors such as digital platform ecosystems should be discussed beyond the context of collectivism. [2] The finding that perceived risk does not significantly impact behavioral intentions in either Japan or China contradicts existing literature on digital health adoption. Therefore, the manuscript should acknowledge this discrepancy and provide a discussion on why risk does not appear to be a factor in this context. [3] The recommendations offered lack specificity regarding actionable steps to enhance mHealth adoption in both Japan and China.
Author Response
Comment 1: The interpretation of cultural differences overly relies on Hofstede’s Individualism-Collectivism framework and overlooks other important cultural and policy-related factors. It would be beneficial to explore Japan’s lower mHealth adoption rates beyond individualism, including aspects such as trust in the medical system or strict privacy regulations. Similarly, for China’s higher mHealth adoption, factors such as digital platform ecosystems should be discussed beyond the context of collectivism.
Response 1: Thank you for your valuable suggestion. We have addressed cultural influences beyond the individualism-collectivism framework in the Discussion section. It includes Japan’s trust in medical institutions and strict privacy regulations as factors influencing lower adoption, while also highlighting China’s robust digital health ecosystem as a driver of higher adoption rates
Comment 2: The finding that perceived risk does not significantly impact behavioral intentions in either Japan or China contradicts existing literature on digital health adoption. Therefore, the manuscript should acknowledge this discrepancy and provide a discussion on why risk does not appear to be a factor in this context.
Response 2: We agree with this recommendation. We have included a discussion on the contradiction between our findings and existing literature in the discussion section, explaining increased normalization of digital health technologies and cultural differences in risk perception influenced by government-backed initiatives and privacy laws.
Comment 3: The recommendations offered lack specificity regarding actionable steps to enhance mHealth adoption in both Japan and China.
Response 3: We have revised the recommendations section to include privacy assurance for Japan, digital platform use in China, and economic incentives to drive adoption.